# An Overview of Research Opportunities to Increase the Impact of Nutrition Intervention Research in Early Childhood and Education Care Settings According to the RE-AIM Framework

**DOI:** 10.3390/ijerph18052745

**Published:** 2021-03-08

**Authors:** Sze Lin Yoong, Jannah Jones, Nicole Pearson, Taren Swindle, Courtney Barnes, Tessa Delaney, Melanie Lum, Rebecca Golley, Louisa Matwiejczyk, Bridget Kelly, Erin Kerr, Penelope Love, Emma Esdaile, Dianne Ward, Alice Grady

**Affiliations:** 1Hunter New England Population Health, Wallsend, NSW 2287, Australia; Jannah.Jones@health.nsw.gov.au (J.J.); Nicole.Pearson@health.nsw.gov.au (N.P.); Courtney.Barnes@health.nsw.gov.au (C.B.); Tessa.Delaney@health.nsw.gov.au (T.D.); Melanie.Lum@health.nsw.gov.au (M.L.); Alice.Grady@health.nsw.gov.au (A.G.); 2School of Health Science, Swinburne University of Technology, Melbourne, VIC 3122, Australia; 3Priority Research Centre for Health Behaviour, The University of Newcastle, Callaghan, NSW 2308, Australia; 4Department of Family and Preventive Medicine, University of Arkansas for Medical Sciences, Little Rock, AR 72205, USA; TSwindle@uams.edu; 5Caring Futures Institute, College of Nursing and Health Sciences, Flinders University, Adelaide, SA 5042, Australia; rebecca.golley@flinders.edu.au (R.G.); louisa.matwiejczyk@flinders.edu.au (L.M.); 6Early Start, School of Health and Society, University of Wollongong, Wollongong, NSW 2522, Australia; bkelly@uow.edu.au (B.K.); emk833@uowmail.edu.au (E.K.); 7Institute for Physical Activity and Nutrition, Deakin University, Geelong, VIC 3217, Australia; penny.love@deakin.edu.au; 8Prevention Research Collaboration, Charles Perkins Centre, Sydney School of Public Health, Faculty of Medicine and Health, The University of Sydney, NSW 2006, Australia; emma.esdaile@sydney.edu.au; 9Department of Nutrition, Gillings School of Global Public Health, University of North Carolina, Chapel Hill, CA 27516, USA; dsward@email.unc.edu; 10Center for Health Promotion and Disease Prevention, University of North Carolina, Chapel Hill, CA 27514, USA

**Keywords:** nutrition, family day care, intervention, implementation science, ECEC, child day care centres, RE-AIM, public health

## Abstract

**Objective:** To highlight opportunities for future nutrition intervention research within early childhood and education care (ECEC) settings, with a focus on generating evidence that has applicability to real-world policy and practice. **Methods:** An overview of opportunities to progress the field was developed by the authors using a collaborative writing approach and informed by recent research in the field. The group developed a list of recommendations aligned with the reach, effectiveness, adoption, implementation and maintenance (RE-AIM) framework. Pairs of authors drafted individual sections of the manuscript, which were then reviewed by a separate pair. The first and senior author consolidated all sections of the manuscript and sought critical input on the draft iterations of the manuscript. **Results:** Interventions that employ digital platforms (reach) in ECEC settings, as well as research in the family day care setting (effectiveness) were identified as areas of opportunities. Research understanding the determinants of and effective strategies for dissemination (adoption), the implementation of nutrition programs, in addition to de-implementation (implementation) of inappropriate nutrition practices, is warranted. For maintenance, there is a need to better understand sustainability and the sustainment of interventions, in addition to undertaking policy-relevant research. **Conclusions:** The ECEC setting is prime for innovative and practical nutrition intervention research.

## 1. Introduction

Poor dietary behaviours, including the regular consumption of foods high in saturated fat, sodium and added sugars, increase the risk of non-communicable diseases such as cardiovascular disease, type II diabetes and some types of cancer [1]. The early childhood years have been identified by the World Health Organization as an important stage for establishing good nutrition and maximising child development and growth [2]. Further, as dietary behaviours are established at an early age, which then continue into adulthood [3], it is particularly important to promote healthy eating behaviours in young children for the prevention of chronic disease. Internationally however, few children are consuming diets consistent with dietary guidelines [4,5]. To improve child diet and reduce dietary-related health risks within the community, leading health organisations [6] recommend the implementation of settings-based population health interventions targeting healthy eating.

Formal early childhood education and care (ECEC) services include centre-based services (such as preschools and long day care) and family day care (also known as family child care homes) [7]. Centre-based ECEC services generally have multiple educators caring for groups of children in a licensed service, while in family day care, individual educators care for a smaller number (on average 10 or less) of children within their own home [7]. ECEC services represent a recommended setting to deliver healthy eating interventions. Globally, these services reach 87% of children aged 3–5 years for up to 30 h per week [7]. Efforts to improve the overall health and nutrition of young children are typically consistent with the overarching philosophy of the sector that applies a holistic approach to child education and learning and considers factors related to physical and social domains, in which healthy eating is one component [8]. This is reflected in both international accreditation standards [9] and guidelines, which recommend that ECEC services create environments supportive of healthy eating behaviours [10,11].

In the last decade, there has been a proliferation of research describing the effectiveness of healthy eating interventions in ECEC services. A 2018 umbrella review of systematic reviews undertaken by Matwiejczyk and colleagues found 12 high quality systematic reviews that summarised the impact of healthy eating interventions in the setting [12]. This review concluded that multi-component and multi-level interventions targeting both environmental-level and individual-level determinants of healthy eating behaviours had a positive effect on the dietary intakes and food choices of 2–5-year-old children in centre-based childcare settings [12]. Other reviews have described several characteristics of interventions associated with improved child dietary intake [13,14]. These include delivery by researchers/external experts [15], the provision of nutrition education programs, training to facilitate peer and educator role modelling [16], interactive, play-based games, providing activities that promote positivity and increasing exposure to healthy eating [17], and increasing parental involvement. Although promising, such interventions may be challenging for implementation in practice. This may be due to the ongoing delivery costs associated with such interventions, the complexity of the interventions, limited adaptability, and interventions which are often incongruent with existing ECEC priorities [18]. For example, many effective interventions include the face-to-face delivery of nutrition education programs by trained experts. Such programs are costly and cannot continue to be delivered with similar fidelity once the research team has withdrawn support or once the program is scaled up to a large number of childcare services. Unsurprisingly, a recent review reported that efficacious obesity prevention interventions no longer retained their effect once they were scaled up in the population [19].

Such findings regarding the lack of impact of health promotion programs at scale are a concern given the significant government investment internationally to support the implementation of nutrition programs into the population [20]. There is an urgent need to better consider how to increase the policy and practice impact of existing nutrition interventions for the sector to ensure such investment produces the intended health outcomes. Given the growth in ECEC-based nutrition intervention research, documented challenges with the implementation of evidence-based programs in the sector, [18] and that many governments are seeking to invest in health promotion programs in the sector, it may be timely to reflect on existing research and consider how to best progress the field to maximise the public health impact of ECEC-based nutrition interventions. The reach, effectiveness, adoption, implementation and maintenance (RE-AIM) framework was developed to address limitations with the translation of health promotion programs, by describing broad dimensions that should be considered to facilitate a broad and equitable population-based impact [21]. In this manuscript, we apply the RE-AIM framework to support the identification and description of research opportunities to improve child nutrition interventions delivered within ECEC settings. By applying this framework, we sought to shift the focus of intervention research from focusing primarily on effectiveness to better considering other aspects of research translation. This paper uses findings from recent relevant systematic reviews and research in the sector known to the authors to highlight opportunities for future research, with a focus on progressing the field of research and generating evidence that has applicability to real-world international policy and practice.

## 2. Methods

### 2.1. Identifying Opportunities

The authors of this manuscript identified potential opportunities to advance nutrition intervention research in the ECEC setting. These authors consisted of researchers and health practitioners with extensive experience conducting research to improve the diets of young children attending ECEC settings in Australia and the United States (two countries where a significant amount of primary research has been generated) [18]. Together, the group, led by the first author (S.L.Y.), developed an overview of opportunities to advance the research using: (a) findings from contemporary, high quality reviews; (b) authors’ and other relevant published research; and (c) experiences undertaking research and practice in the setting. A literature review was undertaken to identify recent, high quality reviews examining nutrition interventions in the setting. Briefly, an electronic search was undertaken in Medline using search terms previously applied in other reviews or validated search filters where available [18,22]: “systematic reviews”, and “early childhood education and care centres” and “randomized controlled trials” and “nutrition”. One reviewer (S.L.Y.) screened the title of all reviews and selected high quality reviews relevant to the scope of this study. The purpose of the review was not to provide a comprehensive overview of the literature but to elicit discussion around future research avenues for the field. Findings from relevant reviews were presented to the author team to inform the generation of research recommendations and support the drafting of each identified area. Initially, one author (S.L.Y.) facilitated multiple small group discussions surrounding each of the recommendations. This was summarised in a written document collaboratively drafted by the authors (see below) and three rounds of iterative comments were sought to revise each of the recommendations. In the generation of research recommendations, we focused specifically on areas to increase the public health impact of ECEC research guided by the RE-AIM framework [21]. Three authors (S.L.Y., J.J. and A.G.) then mapped the identified opportunities to the relevant RE-AIM constructs. The RE-AIM framework has been used previously to organise reviews of health promotion literature, and developers suggest that, together, the specified constructs within the framework be applied to determine the public health impact of an intervention [21]. This activity resulted in seven recommendations outlined in Table 1 and described in the results. In addition to opportunities for research informed by the RE-AIM framework, we also discussed cross-cutting study methodologies that could be applied to generate relevant evidence.

### 2.2. Writing Process

This was an inclusive, group writing process where pairs of authors (typically consisting of a PhD student and a postdoctoral researcher/practitioner) were tasked with identifying and drafting manuscript sections outlining research opportunities and methodologies aligned to their expertise and experiences. A second pair of authors then reviewed the drafted opportunities (i.e., manuscript sections that they did not draft) to provide a broader perspective to the issues discussed. Finally, the group collectively reviewed the entire manuscript to ensure all relevant ideas were captured and included in the synthesis. Two authors (S.L.Y., A.G.) solicited and addressed collective comments and revised the manuscript to ensure the consistency of language and ideas. Such collaborative writing processes provide an opportunity for the contribution of all authors of a manuscript, bring together experiences of other researchers and practitioners, enabling ideas to be solicited more broadly, and allowing for the capacity building of both early- and mid-career researchers [23].

## 3. Results

### 3.1. Reach and Effectiveness

#### 3.1.1. Assessing the Efficacy of Digital Health Interventions Targeting Child Nutrition and Service Environment

Digital platforms (e.g., telehealth, web, apps) are an attractive medium for the delivery of nutrition interventions, as most ECEC services have the existing infrastructure (computer and internet access) [24,25,26] and find these platforms acceptable to support the implementation of healthy eating policies and practices [24,26]. Importantly, digital interventions can typically be delivered with high fidelity and at low cost [27], providing an opportunity for wide-scale implementation.

The shift towards the use of digital technologies to support the achievement of health and education objectives has been expedited since the global COVID-19 pandemic [28,29,30]. In ECEC, many traditional models of face-to-face support have been replaced by remote, technology driven support modalities, such as Skype, Zoom and email (internal communication New South Wales (NSW) Ministry of Health). With the continuing challenges to research now exacerbated by the COVID-19 pandemic, digital health interventions represent a promising way forward for future nutrition research in the setting. Findings from a national survey with Australian ECEC services (conducted by Grady, Yoong, Barnes), found 76% of services would like support (e.g., email, telephone, text) to use digital health interventions to improve nutrition (unpublished data from the authors).

Previous reviews [18] have identified only one existing randomised controlled trial (RCT) that examined the use of a web-based intervention to deliver a nutrition program in the setting [31]. There is further emerging evidence, however, of the feasibility, acceptability and efficacy of digital health interventions in this setting [24,32,33]. To our knowledge, only two published RCTs evaluating the impact of digital interventions for healthy eating in ECEC exist [31,32,34]. The first, a six-month pilot RCT in the United States (US) of a web-based nutrition practice support program (Go NAPSACC), found improvements in the nutrition environment of ECECs allocated to the intervention group compared with the control (albeit, not statistically significant) [31]. The second, a 12-month RCT of a web-based menu planning program conducted across NSW, Australia, demonstrated significant increases in the provision and consumption of some healthy foods, and a significant reduction in the consumption of unhealthy (discretionary) foods [32,34], and reported cost savings compared to usual care [27].

Although promising, the effect and ongoing impact of digital interventions remain largely unknown. Both RCTs described above found variable engagement with digital health interventions and attenuating exposure—both key drivers of behaviour change [35]. Such findings suggest that a greater consideration of the barriers and enablers to the adoption of digital health interventions in the setting [32] is needed, as well as investment in formative evaluations of such technologies. Further, there is an opportunity for the use of adaptive designs to understand the effective components of digital interventions. Additionally, designing digital health interventions that can be embedded into existing structures within ECEC [36,37] or are closely aligned with reporting requirements for the sector [38] may be a way of delivering nutrition interventions, while addressing potential barriers to engagement with such technology.

#### 3.1.2. Building the Evidence of Effective Nutrition Interventions within Non-Centre-Based ECEC Services

The majority of the research described in this manuscript has been undertaken in centre-based ECEC, which mainly include long day care and preschools [12,18]. The other primary area of care within ECEC (termed as non-centre-based) is family day care [7]. Approximately 8% of children who attend childcare in Australia [39], and 11% (almost 1 million) of children in the US, attend family day care [40,41]. Such services may cater to a larger proportion of children from lower socioeconomic status backgrounds given the overall lower daily fees for this service [42].

Compared to centre-based ECEC settings, limited nutrition intervention research has been conducted in family day care services [43,44]. A review of the obesogenic characteristics of the family day care environment in the US highlighted a need for more comprehensive policies and professional development opportunities, focusing on reducing controlling feeding practices, reducing the provision of discretionary foods (fried foods and sweetened beverages), communicating with families, dispelling educators’ false beliefs and changing perceptions related to feeding practices [43]. These findings have been supported by studies from Australia [45,46], Canada [47] and the UK [48], highlighting the opportunities for nutrition interventions in this setting.

A systematic review of interventions conducted up to March 2019, however, found no RCTs aiming to improve the diet, physical activity and/or weight of children aged 0–6 years in family day care services [44]. Only two controlled trials were identified, Romp and Chomp [49] and the Healthy Kansas Kids program [50], which both reported on nutrition-related outcomes using self-report environmental surveys. Intervention strategies included in both interventions were educational meetings, outreach visits and materials. Improvements were found in offering fruits and vegetables [50], the availability of healthier options on menus [50], the provision of nutrition education [50], fewer unhealthy food items offered [49] and educator practices that support positive meal experiences [49,50].

There is a clear gap for primary research in the family day care setting to identify interventions that are effective in improving children’s diets. Whilst interventions among centre-based childcare could presumably be delivered in the family day care setting, several characteristics of this setting, including the smaller numbers and wider age range of children, the single-carer environment and the differences in physical space and in operating structures as well as available resources, necessitate strategies that are tailored to the needs of the setting [51]. For example, since the conduct of the systematic review [44], the first cluster RCT in the setting, the Keys to Healthy Family Child Care Homes, was undertaken [52]. To address the unique challenges faced by family day care providers, a tailored intervention model was developed, focusing on educator health, family day care nutrition environment and business practices to support providers delivering healthy eating practices. Significant improvements in children’s diet quality, increased consumption of wholegrains and seafood/plant-based protein, and reduced intake of refined grains and sodium [52] were documented, suggesting the benefits of such an intervention in the setting. Further controlled trials examining innovative interventions that target family day care-specific obesogenic factors are needed to improve the diet of children who attend family day care services. Investment in such research may assist in ensuring adequate reach and equity of access to evidence-based ECEC nutrition programs.

### 3.2. Adoption

#### Identifying Determinants and Strategies to Increase the Dissemination of Nutrition Interventions, Programs and Guidelines

Dissemination research seeks to increase the targeted distribution of health interventions, programs and guidelines to an identified targeted audience to increase the intention to use and adopt the targeted program [53]. Dissemination is an essential prerequisite of implementation. Therefore, an understanding of effective dissemination processes is essential to increase the scale-up and adoption of programs, particularly for interventions where limited ongoing implementation efforts are needed once the decision to adopt has been made [53]. For example, for technology-based interventions that require little implementation by an end-user (e.g., app-based communication with parents to pack healthier lunchboxes controlled centrally) or environmental interventions (e.g., restructuring the eating environment, nudge strategies to increase selection of healthier foods), efforts to understand how to increase the adoption of the program at a population level is critical. The identification of characteristics of interventions which require little implementation effort, and the improved understanding of determinants (barriers and enablers) to the adoption of such nutrition interventions across different jurisdictions are needed to design effective dissemination strategies to increase reach, and therefore impact, of these interventions.

To our knowledge, there have been few explorations of the barriers and enablers to the dissemination of nutrition programs or guidelines. Further, there have been a limited number of controlled evaluations that have assessed the impact of dissemination strategies specifically in the ECEC setting. We are aware of one national study with 407 Australian ECEC services that explored barriers and enablers to the adoption of digital health interventions to improve ECEC nutrition environments using a technology-specific dissemination framework [54]. This study found that frequently reported barriers among ECEC providers related to team interactions and the organisations’ capacity to innovate. Enablers related to increasing the ease of adoption decisions and identifying work and individuals involved in taking up the new innovation [54].

A Cochrane systematic review assessing strategies to improve the implementation of nutrition and physical activity programs in ECEC settings found that none of the 21 studies reported the impact of dissemination strategies on the adoption of evidence-based policies, programs and practice [18]. We are aware of two controlled trials, one assessing the impact of strategies to increase the adoption of dietary guidelines and the other an online menu planning intervention. The first was a RCT with 77 NSW-based childcare services, which found that educational material informed by the theory of planned behaviour increased ECEC services’ intentions to adopt sector-specific nutrition guidelines [55]. The second, a national controlled trial with 46 ECEC services across Australia, found that providing training, telephone contact and resources increased services’ adoption of an online menu planning program compared to a single email notifying of access to the program [56]. There is a clear opportunity to better understand the determinants of dissemination, as well as strategies to facilitate the dissemination of nutrition interventions and guidelines at scale, across different jurisdictions in the ECEC setting.

### 3.3. Implementation

#### 3.3.1. Identifying Determinants and Strategies to Increase Implementation of Nutrition Interventions, Programs and Guidelines with Fidelity

Implementation research seeks to identify effective strategies to integrate evidence-based policies and practices within end-users settings [57], including ECEC. The failure to implement effective nutrition interventions, programs and guidelines with fidelity in this setting has been well documented [58,59,60]. In parallel with efforts to identify effective interventions, research has been conducted increasingly in the last decade to identify strategies to increase the implementation of nutrition programs in the sector as intended.

Within a recent Cochrane review described above [18], almost all of the 21 included studies tested multi-component implementation interventions, with meta-analyses indicating that studies were effective in improving the implementation of nutrition and physical activity policies and practices (nutrition practices were not reported separately) (SMD 0.49; 95%CI 0.19, 0.79 and OR 1.83; 95%CI 0.81, 4.11) [18]. The findings across the studies were highly variable, and the certainty of the evidence was deemed to be low–moderate [18]. The review found that the included studies employed a narrow range of implementation strategies, such as educational materials and educational meetings, which if employed in isolation were unlikely to be effective [18]. Further, due to the heterogeneity of the interventions and outcomes employed across studies, the review authors were unable to isolate the effects of specific strategies (or combinations) on the implementation of policies and practices [18]. The review highlighted a number of limitations to the evidence-base, including small sample sizes, use of self-reported measures, lack of economic evaluations of nutrition interventions, and a lack of theory to guide the selection and development of strategies to address the determinants of implementation. Given this, the current literature provides limited guidance to support the selection of strategies to improve the implementation of nutrition interventions within the ECEC setting.

Future research using comprehensive theoretical frameworks [61] to allow for the identification of factors impeding implementation, and then to guide the selection of implementation strategies accordingly, may help with increasing the impact of implementation interventions [18]. Employing such frameworks can assist in ensuring that contextually relevant strategies are employed within interventions and maximise the likelihood of achieving improvements in the implementation of evidence-based nutrition policies and practices within the ECEC setting [62]. Further, there is a need to disentangle and measure the effectiveness of individual implementation strategies within multi-component interventions and to describe whether these strategies targeted the identified barriers to implementation. This is a challenging process, as the determinants of policy and practice implementation are complex, and the mechanisms by which these strategies facilitate implementation are not well understood [63]. There is an opportunity to apply theoretical frameworks and better articulate logic models to understand the mechanisms of implementation [64] and employ innovative designs such as a multiphase optimization strategy (MOST) [65] to disentangle the effects of the implementation strategies. Authors of this manuscript are currently undertaking a study using the MOST design to examine the impact of a multicomponent ECEC-based intervention targeting food service, mealtime environment and curriculum to improve vegetable consumption [66]. Additionally, using consistent taxonomy to describe implementation strategies including the expert recommendations for implementation change (ERIC) taxonomy [67] or behaviour change techniques [68] provides an opportunity to disintegrate the interventions into more granular components and this may allow for the exploration of differential effects using meta-regression techniques similar to those carried out in previous systematic reviews [69].

#### 3.3.2. Identifying Determinants and Strategies for De-Implementation of Inappropriate Nutrition Practices

In contrast to the growing evidence around implementation, there has been limited research addressing the de-implementation of inappropriate nutrition practices. De-implementation has been described as stopping, reducing or replacing the use or delivery of practices or services that are unproven, harmful, ineffective or inappropriate [70]. Certain educator mealtime nutrition practices used within the ECEC setting have been associated with negative impacts on child eating behaviours, such as the impaired ability to self-regulate food intake [71] and poor child dietary outcomes [72]. Such practices include the use of controlling, non-responsive feeding behaviours, including pressuring children to eat [72], withholding of palatable foods [73] and using certain foods as a reward for eating or to encourage other desirable behaviours [72]. Supportive interventions to assist educators in reducing or stopping such practices may create the space for replacement with evidence-based feeding practices [74].

To our knowledge, however, there are currently no published trials with an explicit focus on the de-implementation of inappropriate nutrition practices in ECEC settings. Several trials targeting the improvement of ECEC nutrition practices have included both practices to be implemented (e.g., praise children for trying new foods, role model eating healthy foods) as well as de-implemented (e.g., requiring children to sit at the table until they clean their plates, using food to encourage appropriate behaviour) [31,75,76,77,78,79,80,81]. Many of such studies have reported promising results on the implementation [81,82], however, without a distinct de-implementation lens, little can be concluded about effective de-implementation strategies from such trials [83]. Specifically, trials that consider de-implementation when designing interventions (i.e., when choosing theories, strategies and strategy mechanisms), differentiate de-implementation aims from any implementation aims, and measure outcomes in the context of de-implementation, are required [83].

At present, Swindle and colleagues are testing a multi-faceted de-implementation strategy that leverages a peer learning collaborative with classroom-based goal setting around feeding practices, which educators select to “stop” and “start” [84]. This work, combined with prior studies on the determinants of nutrition practices in ECEC [72,73,85] can inform further research into identifying effective de-implementation strategies for this setting. Specifically, future research is required to determine how de-implementation processes differ from implementation in ECEC, the differential mechanisms to target for de-implementation (compared to implementation), and the long-term sustainability of de-implementation efforts.

### 3.4. Maintenance

#### 3.4.1. Understanding Sustainability and Sustainment of Nutrition Programs

To maximise the public health impact, effective nutrition interventions must be implemented in an ongoing, sustained manner. Sustainability has been defined by Scheirer and Dearing as the “continued use of a program for the continued achievement of desirable program and population outcomes” [86].

Considering the sustainability of nutrition interventions in the ECEC setting (and more broadly) is important, as improvements in nutrition-related health outcomes (such as body mass index and biochemical markers) often take time to accrue [87]. Additionally, reviews suggest that when sustained implementation is not achieved, relationships with external stakeholders can be compromised [88], and the prevalence of implementation can regress to baseline levels [89]. Planning for sustainability requires the consideration of ongoing funding and resources as well as the potential loss of investment in the event that effective interventions are not sustained [87].

In the ECEC setting, the recent Cochrane systematic review found that, collectively, studies were effective in improving implementation [18]. However, no studies reported on the sustainability of the interventions tested [18]. In addition, just six studies embedded long-term measures of follow-up (≥3 months post-intervention), with the majority measuring intervention outcomes immediately post-intervention [18]. Further, the duration of intervention delivery within included studies was rarely >12 months, limiting the opportunity to assess sustainability. As such, the degree to which nutrition interventions are sustained in ECEC remain largely unknown.

Despite the clear importance, there is little evidence regarding how best to support the sustained implementation of effective nutrition interventions in ECEC. Previous studies have reported common impediments to nutrition intervention sustainment in ECEC to include high staff turn-over, insufficient resources to support ongoing implementation, parental engagement, and lack of prioritisation of programs in an environment of limited resources [90]. A comprehensive understanding of the determinants (factors) of sustainment of nutrition interventions in ECEC is needed to guide the development of future initiatives. The use of sustainability-specific, theoretically informed approaches to guide the planning, implementation, and evaluation of interventions has been recommended to improve the sustainability of public health interventions [87]. Further, embedding long-term follow-up measures in ECEC nutrition trials is recommended to assist in understanding the intervention implementation over time and possible attenuation [87].

#### 3.4.2. Undertaking Policy and Regulatory Relevant Research

Health policies and regulatory frameworks are important tools for governments to support sustained improvements to public health by outlining a set of actions and recommendations to achieve particular health goals [91]. In the past decade, international organisations, such as the World Health Organization, have introduced ECEC-relevant frameworks that provide broad recommendations to create supportive nutrition environments to improve child diet [92]. Such recommendations have been adopted in many countries as part of national and regional strategies targeting obesity prevention in young children [92]. A policy mapping exercise of obesity prevention policies in England and Scotland, Canada, the Republic of Ireland, New Zealand and Australia found that many countries had obesity prevention policies that included recommendations targeting the nutrition behaviours of children attending ECEC settings [92]. These include setting standards for child diet, having service-level policies and procedures to support meeting standards, and training programs to capacity build ECEC staff.

Despite the availability of such broader policies, mandatory and detailed guidance which define healthy eating are not commonplace, potentially compromising the optimal food provision [92]. Many previous nutrition interventions have focused on supporting ECEC settings to establish service policies; however, without clear guidance from broader frameworks, such policies are likely to have a limited impact in improving dietary guideline adherence [93]. Additional complexity also exists where there are regional differences in state/region regulation that do not necessarily align with national policies. In Australia, for example, ECEC service accreditation processes are tied to existing regulatory frameworks federally [9]. However, state and territory agencies are responsible for ensuring compliance [9,39]. The assessment and monitoring of compliance is further exacerbated by differences in the interpretation of federal nutrition guidelines by states and territories [94]. This discrepancy between levels of government is similar in the US, with “Caring for Our Children” detailing the National Health and Safety Performance Standards for ECEC in the US [95], yet licencing of ECEC services is determined at a state-level with accreditation of ECEC services being optional. A better understanding of the similarities and differences between these policies and whether they map to empirical evidence may be needed to support efforts to generate clearer recommendations.

Although clearer guidance at the regulatory level is needed to support consistent messages and sustained changes in ECEC-based nutrition practices, there is an absence of empirical research to support these decisions. Research at this broader level is highly challenging, necessitating strong partnerships between research agencies, national- and state-level policy makers and practitioners and may require the application of alternative empirical designs such as interrupted time series, similar to that employed in alcohol research [96]. Monitoring systems such as INFORMAS (International Network for Food and Obesity/non-communicable disease) [97], which monitor both the service and regulatory environments of ECEC settings, provide important and unique opportunities to explore such policy-relevant questions. Additionally, efforts to understand the decision-making processes and priorities of policy makers to enable the design of policy-relevant research are crucial to inform the development of evidence-based regulatory policies for the sector.

### 3.5. Study Methodology to Advance Translational Research

#### 3.5.1. Use of Qualitative and Mixed Methods Approaches to Design and Evaluate Nutrition Interventions

Previous evaluations of nutrition interventions in this setting have largely used quantitative approaches. Although such approaches are critical to describe the impact of an intervention, they provide limited insight into the potential reasons for the success or failure of nutrition interventions in ECEC, nor the usefulness of implementation-related outcomes. Qualitative or mixed methods research is needed to understand why the translation of evidence-based nutrition practices into daily routines by childcare staff is difficult to achieve [12]. Using a socio-ecological framework and predominately semi-structured interviews, researchers have identified that it is a reciprocal and complex interplay of factors delivered through structures (e.g., meal provision) and processes (i.e., the interactions and activities that occur in services) at the individual, service and societal level of influence that determines nutrition-related practices and healthy eating behaviours in children [57,98,99,100,101,102]. Findings from systematic reviews concur that the most effective interventions are multi-strategy, system-wide approaches which focus on each level of influence [12]. Further qualitative research exploring ECEC staff experiences and perceptions provides valuable insights at each level of influence, such as changes in nutrition knowledge, and practices and beliefs [103,104] as well as issues of accessibility, availability and affordability that may not be identified if not explored via qualitative research methods. Similarly, at the service level, healthy eating policies and government-developed resources that dictate nutrition practices are viewed as not meeting the needs of childcare staff, constraining staff autonomy and restricting practices promoting children to make healthy food choices [99,100,102].

More recently, mixed methodology has been used to gain further understandings from nutrition trials undertaken in ECEC services. Swindle and colleagues undertook a mixed-method analysis with early childhood educators to understand the barriers and facilitators to the implementation fidelity of a nutrition intervention in ECEC services in the US [105]. The study employed semi-structured interviews and a directed content analysis approach informed by the integrated Promoting Action on Research Implementation in Health Service framework [105,106]. A range of relevant constructs including contextual factors (i.e., culture, leadership support, mechanisms for embedding change), recipient characteristics (i.e., beliefs about what works, personalized strategies to use the intervention), innovation (i.e., time, preparation, degree of fit, intervention advantage), and facilitation (i.e., trainer support, desire for additional training) were identified. As a result, new strategies to target such constructs to improve the implementation fidelity were suggested that would not have been identified using quantitative evaluation alone.

As the evidence base continues to grow, qualitative and mixed methods evaluations that explore not only program efficacy, but allow for the exploration of factors to support and assess translation into practice are key to increase the public health impact of nutrition research within the sector.

#### 3.5.2. Hybrid Trial Designs

In addition to mixed methods evaluation, hybrid trial designs, which blend design components of both intervention effectiveness and implementation trials [107] are a potential way of providing research evidence on both the efficacy of an intervention as well as considering factors relating to real-world translation [55]. Three types of hybrid designs are described in the literature, which vary in terms of their emphasis on intervention or implementation outcomes [107].

In hybrid type 1 designs, the primary aim is to test the intervention effectiveness, whilst the secondary aim is often to gather data to inform future potential implementation [108]. Hybrid type 2 designs place a more equal emphasis on testing both the intervention effectiveness and the implementation strategy (as co-primary aims) [107]. Finally, hybrid type 3 designs focus primarily on the effectiveness of the implementation strategies via measuring outcomes such as the fidelity and adoption, with the secondary aim of gathering data on the intervention effectiveness [107,108].

The current extent of use of hybrid trial designs in ECEC nutrition research is not fully known, as trials applying such designs may not necessarily be defined as such given the terminology has only been recently coined [107]. In the recent Cochrane review of ECEC nutrition, physical activity and obesity implementation trials, und nine of the 21 studies (four specific to nutrition) could be considered to have employed a hybrid design (due to the inclusion of both intervention and implementation outcomes), although none were specifically described as such [18]. However, since the publication of this review, authors of this manuscript (Yoong, Barnes, Pearson, Grady) have published a type 3 hybrid trial primarily focused on assessing the impact of a web-based implementation strategy in increasing ECEC service adherence to dietary guideline recommendations [32] while assessing the impact of the strategy on child dietary intake in a nested sample of children [34]. The web-based implementation strategy was effective in increasing the provision of healthier foods and improving child diet, and also offered substantial cost-savings to usual care [27]. The web-based strategy received federal funding for roll out nationally in Australia, in part due to the capacity of such hybrid trials to provide relevant information to support decision making.

In addition to this trial, at least three recent protocols specifying the use of hybrid designs for ECEC nutrition research have been published [38,104]. Firstly, a protocol by Barnes and colleagues describes a hybrid type 2 design, which aims to pilot the feasibility of a web-based intervention as a strategy to improve the implementation of recommended nutrition policies and practices while also testing the effectiveness of the intervention on child dietary intake [38]. The remaining two protocols describe hybrid type 3 trials. The first trial by Ward and colleagues aims to compare an enhanced versus basic version of the GoNAPSACC online program to increase the service’s adoption, implementation and maintenance of service nutrition and physical activity practices and collects data on child diet intake [109]. Swindle and colleagues’ hybrid type 3 protocol compares an enhanced implementation strategy to a usual implementation strategy for a multicomponent obesity prevention intervention in childcare services [104], whilst also gathering data on child health outcomes (child fruit and vegetable intake and body mass index) [105].

Whilst evidence of increasing hybrid trial design publications is promising, the greater consideration and routine use of hybrid designs is recommended for their potential to expedite the translation of evidence into practice and maximise the impact of ECEC nutrition interventions [109]. Researchers currently employing such dual-purpose designs can benefit from the increasingly available literature, defining and describing hybrid design methodology and reporting, allowing for a greater consistency in the use of terminology and the easier identification of trials [107,110,111,112].

#### 3.5.3. Applying a Health Equity Lens to the Design of Nutrition Interventions

The ECEC environment reaches children across various socioeconomic and demographic groups, making it a prime context for the consideration of interventions and implementation approaches that advance health equity. However, there is a limited explicit application of a health equity lens in the design of nutrition interventions in ECEC, despite their potential to reach minority and low-income populations. Intervention and implementation research with a healthy equity lens seeks to understand and address determinants of disparate health outcomes across groups [113]. Specifically, interventions aimed to promote health equity explicitly considers behavioural, sociocultural, biological, physical environment, and systems factors that contribute to health disparities across multiple levels of influence (individual, interpersonal, community, and societal) [114]. This can require interventions that are “complex, comprehensive, and realistic to life conditions (p. 511)” [115]. Further, all health interventions delivered in this setting need to consider the systemic and economic context driving the workforce capacity of the setting, which is likely to be exacerbated in low resource services. Many studies have documented that ECEC educators in the US tend to have lower income, are more likely to be enrolled in public health support programs, are disproportionately women of colour and have poorer physical health [116]. Interventions that fail to consider this complexity and the different contexts can inadvertently contribute to the maintenance and/or exacerbation of health disparities. For example, an intervention or implementation strategy that is only sustainable in ECEC services with high resources would increase, rather than decrease, health disparities. The engagement with local stakeholders, such as co-design approaches, in the development of both nutrition interventions and implementation strategies is important to promote culturally and socially relevant interventions with the best chance of promoting equal access to health [115]. A number of strategies to advance the equity in ECEC-based nutrition research may include, but are not limited to: (a) the design/adaptation of interventions and implementation strategies to address multiple levels of health influence including those less frequently considered (e.g., structural factors), (b) the design of interventions to reach populations with health disparities, (c) the co-design of nutrition interventions, programs and guidelines that are tailored to end-users, and (d) the evaluation of interventions and implementation strategies with rigorous quantitative and qualitative methods to determine if outcomes are equitable across groups [117].

### 3.6. Limitations

The research opportunities outlined here primarily represents the view of the authors on this piece. However, substantial effort was made to identify and draw on findings from relevant reviews and other empirical research to ensure that this was reflective of the broader nutrition literature within ECEC. The first author also actively facilitated broad input from other authors via a group drafting processes to ensure a diverse range of views and an understanding of the literature was represented. The literature search and discussion focused specifically on nutrition research within ECEC, with limited consideration of parallel fields including education and business. Future researchers should consider drawing on the evidence from broader non-health fields to inform the development of nutrition interventions within this setting. Additionally, the views of educators and managers that implement nutrition programs within ECEC were not sought for this manuscript. The author team consists of public health practitioners who deliver nutrition programs to ECEC settings and these perspectives were actively sought to ensure applicability to the setting. While this manuscript provides a broad overview of areas of opportunities for future research, we strongly recommend that all nutrition research undertaken in the setting be co-developed with ECEC staff to ensure the relevance, acceptability and real-world impact of such programs. There are likely many other areas of nutrition research that may be important to progress that were not included in this manuscript as we attempted to prioritise those that were considered more likely to have a policy and practice impact, informed by the RE-AIM framework.

Notwithstanding such limitations, this manuscript provides a comprehensive overview of recent and relevant nutrition research in the ECEC setting, and uses this to highlight innovative areas of focus, as well as intervention approaches that can be applied to advance the field.

## 4. Conclusions

In a short period of time, a large evidence base of nutrition intervention research has been produced, suggesting that multi-level interventions in centre-based childcare settings have positive effects on children’s nutrition. Using a collaborative writing approach amongst researchers and practitioners with expertise in nutrition interventions in ECEC, we propose a number of research opportunities and methodologies to increase the real-world relevance, translation and impact of ECEC-based nutrition interventions, according to the RE-AIM framework. To increase the reach and effectiveness of existing nutrition interventions, we propose interventions that use technology-based platforms as well as more intervention research in the family day care setting. There is an urgent need to understand the determinants of, and how to support, the dissemination and implementation of nutrition programs as well as the de-implementation of practices with little known benefits. Additionally, we have identified gaps in understanding the sustainability of nutrition interventions in ECEC settings and the need to better understand the regulatory context and its impact on ECEC environments. Lastly, we propose a number of study methodologies that could be used to advance the literature, including qualitative and mixed-method designs, hybrid trial designs and applying a healthy equity lens in the design of nutrition interventions in the setting.

## Figures and Tables

**Table 1 ijerph-18-02745-t001:** Opportunities to advance nutrition intervention research in early childhood and education care (ECEC) settings according to the reach, effectiveness, adoption, implementation, maintenance (RE-AIM) framework.

Construct	Reach and Effectiveness (Combined Due to Overlapping Content)	Adoption	Implementation	Maintenance
Research opportunities	Assessing the efficacy of digital health interventions targeting child nutrition and service environment.	Identifying determinants and strategies to increase the dissemination of nutrition interventions, programs and guidelines.	Identifying determinants and strategies to increase the implementation of nutrition interventions, programs and guidelines with fidelity.	Understanding the sustainability and sustainment of nutrition programs.
	Building the evidence of effective nutrition interventions within non-centre-based ECEC services.		Identifying determinants and strategies for the de-implementation of inappropriate nutrition practices.	Undertaking policy and regulatory relevant research.
Study Methodology	Use of qualitative and mixed-method approaches to design and evaluate nutrition interventions. Hybrid trial designs. Applying a health equity lens to the design of nutrition interventions.			
Next steps for research	**Using proposed study methodology, to:**			
	Develop and test scalable interventions amenable for delivery into the population (e.g., digital health intervention, nudge intervention);			
	Conduct controlled trials targeting child nutrition within the family day care and out of school care setting			
	**Use theory-informed methods to assess determinants to dissemination and implementation of evidence-based nutrition interventions, programs and guidelines.**			
	Undertake controlled trials assessing strategies to improve the dissemination and implementation of evidence-based nutrition interventions, programs and guidelines, particularly at scale.			
	Increase the understanding of determinants and strategies to facilitate the de-implementation of evidence-based nutrition interventions, programs and guidelines,			
	Consider sustainability at the design of nutrition interventions and measure the sustainability of interventions where there is an opportunity.			
	Identify opportunities for policy-relevant research and increase the understanding of public health decision-making for the sector.			

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
