# Peer review of "An Overview of Research Opportunities to Increase the Impact of Nutrition Intervention Research in Early Childhood and Education Care Settings According to the RE-AIM Framework"

_ijerph, 2021, doi:10.3390/ijerph18052745_

Round 1

Reviewer 1 Report

The topic of this paper is very interesting. There are some significant issues that need to be addressed prior to publication. How were the searches conducted for each of the sections? What where the inclusion/exclusion criteria? What questions guided the overall research and/or the searches for individual sections of REAIM?  From this, there should be additional context added to the introduction that pertains to those questions, and appropriate text in the methods describing the strategy. 

Reviewer 2 Report

This paper presents a novel look into the existing literature on the ways ECEC environments can promote child health. It is comprehensive, well-written and conceived and provides an important call to other researchers and funders in this field.

One major concern relates to the sources of evidence used to develop the recommendations: how they were identified, reviewed/coded and selected for inclusion. In several domains, it would be helpful to draw from the larger ECE-focused literature (aka not health, but education) on strategies for dissemination and implementation in those settings. While not health- or nutrition-related, that literature could address some of the gaps identified in this review. At a minimum, suggesting that future researchers consult this literature could be added. 

It would also be helpful to acknowledge the limitation that ECEC providers were not included/consulted for this project. Those stakeholders may have been able to shed light on the reach challenges faced by prior studies. You mention "staff motivation" and "resistance to change" as impediments to sustainment (section 3.4.1).  This language fails to recognize the day-to-day experience of the ECEC workforce, who are often low-income women of color with low educational attainment levels and few opportunities for professional advancement. I would suggest revising the language to better acknowledge the challenging environment of these providers, and how that environment and systemic policies (e.g. low wages, minimal benefits) contributes to their "motivation" to engage in additional programming efforts. For example, low motivation could be reframed as "insufficient support/resources provided to ECEC providers to support implementation" or "inconsistent messaging given to providers about which interventions are important to implement" or "insufficient justification given to providers about why these interventions are necessary". This may also fit within the health equity section.

There is inconsistent use of determinant/determinants throughout (e.g. in text, headers and Table 1). 

May want to clarify in section 3.3.1 that you are assessing full/complete implementation with fidelity. There are several other places (e.g. beginning of 3.4.1) where the addition of "..with fidelity" could be added to better clarify intent.

I suggest adding another table summarizing the suggestions for next steps for researchers, or adding some of that detail to Table 1.

Round 2

Reviewer 1 Report

The authors have addressed the highlighted concerns before. While a narrative review, this manuscript does provide relevant information for those who may with to utilize this framework in early child care settings to promote nutrition. 

Reviewer 2 Report

The authors addressed all of my prior comments and the improvements to the paper are notable.